# How Metagenomics Has Transformed Our Understanding of Bacteriophages in Microbiome Research

**DOI:** 10.3390/microorganisms10081671

**Published:** 2022-08-19

**Authors:** Laura K. Inglis, Robert A. Edwards

**Affiliations:** Flinders Accelerator for Microbiome Exploration, College of Science and Engineering, Flinders University, Bedford Park, SA 5042, Australia

**Keywords:** metagenomics, bacteriophage, microbiome

## Abstract

The microbiome is an essential part of most ecosystems. It was originally studied mostly through culturing but relatively few microbes can be cultured, so much of the microbiome was left unexplored. The emergence of metagenomic sequencing techniques changed that and allowed the study of microbiomes from all sorts of habitats. Metagenomic sequencing also allowed for a more thorough exploration of prophages, viruses that integrate into bacterial genomes, and how they benefit their hosts. One issue with using open-access metagenomic data is that sequences added to databases often have little to no metadata to work with, so finding enough sequences can be difficult. Many metagenomes have been manually curated but this is a time-consuming process and relies heavily on the uploader to be accurate and thorough when filling in metadata fields and the curators to be working with the same ontologies. Using algorithms to automatically sort metagenomes based on either the taxonomic profile or the functional profile may be a viable solution to the issues with manually curated metagenomes, but it requires that the algorithm is trained on carefully curated datasets and using the most informative profile possible in order to minimize errors.

## 1. Introduction

The microbiome is the microbial component of an ecosystem. It includes bacteria, archaea, and viruses and is an essential part of most ecosystems. It can greatly influence human and environmental health, and it can be heavily impacted by human activities.

Microbiomes are usually very diverse, and a microbiome dominated by a few species, a dysbiosis, is often a cause or symptom of a disease state [1,2]. Similar to macro-scale food webs, bacteria and viruses operate under complex parasitic and/or symbiotic relationships. Studying these dynamics can provide insight into disease and provide potential solutions for pressing issues such as antimicrobial resistance.

Historical approaches to studying the microbiome relied on in vitro culture as the first step in isolating a workable sample. However, only a small fraction of microbes could be cultivated, leading to a massive underestimation of the diversity of the microbiome. This source of error is known as the great plate count anomaly [3]. Currently, the main approach to studying the microbiome is through metagenomic sequencing, which does not require laboratory culture.

This review will briefly summarise the history of metagenomics and describe microbiome research made possible through the availability of metagenomic sequencing. Then, it will explore potential approaches for categorising metagenomes based on their isolation environment and compare the pros and cons of manual versus algorithmic methods for this research. Finally, the review will consider how metagenomic sequencing has improved research into bacteriophages, particularly for determining how these viruses operate within microbiomes.

## 2. A Brief Overview of Metagenomics

The term metagenome refers to the sum of the genomes found in a tested sample and was first coined in 1998 [4]. Outside of examining the few microbes that could be cultured, the first attempts to examine the wider metagenome were performed by isolating total DNA from fresh samples, then cloning large DNA fragments into plasmids or bacterial artificial chromosomes (BACs) maintained in microbes such as *Escherichia coli*. High-throughput screening was then used to examine the chemical diversity produced by the clones [4] or, later, sequencing the cloned fragments [5]. The first published metagenomes were sampled from acid mine drainage biofilms in early 2000 [6]. Since then, the methods for analysing metagenomes have been consistently improving to the current day. Now, we perform metagenomic sequencing using massively parallel sequencing or deep sequencing. This involves sequencing millions of small fragments of DNA, and then recreating the genome by connecting the fragment sequences using bioinformatics analyses [7].

There are two main methods for microbiome sequencing: amplicon sequencing and whole genome sequencing. Amplicon sequencing detects only the target gene, and usually focuses on the highly conserved 16S gene because of its ubiquity in bacterial genomes. This approach is relatively cheap and simple; however, it is limited in scope: the common 16S gene primers only target some bacteria, and depending on what primers, reference databases, and bioinformatics settings are used different genera can be underrepresented or missed entirely [8]. Viruses, archaea, and Eukarya go undetected by this method. Whole genome sequencing, as the name implies, sequences all the DNA in a sample. This approach has the benefit of being able to detect viruses and has also been shown to be more accurate in detecting bacteria to the species level [9].

Technological improvements have led to metagenomic sequencing becoming cheaper and more accessible, and it is now being used in many areas of research, from environmental to medical sciences. Since metagenomic sequencing was first applied to diagnose infection in a human patient in 2014 [10], its use as a clinical tool has slowly increased [11,12]. The fact that metagenomic sequencing targets all a sample’s genetic material at once means the approach has many advantages over traditional diagnostic methods, which are generally limited to microbes that are well studied [13,14,15,16,17,18] and to testing for only a couple of potential pathogens at a time.

Metagenomic sequencing is also useful for monitoring pathogens in non-human settings. Recently it has been applied to monitor sewage for the SARS-CoV-2 virus [19,20,21], and has proved to be an effective monitoring tool, detecting SARS-CoV-2 in a community before any patients tested positive [22]. Metagenomic sequencing is also applied more broadly to monitor pathogens in human communities, including both bacterial and viral pathogens [23], and to examine or monitor antibiotic resistance genes [18,24].

The wider microbiome is also explored with metagenomics [1,25,26]. Only 1% of microbes can be cultured [2], limiting our capacity to understand the entire microbiome using traditional methods. Metagenomics has enabled access to the remaining 99% that was almost entirely unknown to us 20 years ago [25] and may be used to quickly screen microbiomes for useful functions [27]. For example, these previously uncharacterized microbes could be important indicators of host health [1,25,26] or reveal opportunities to develop new antimicrobials [2,28,29].

The Human Microbiome Project began with the intention of documenting the microbiome of healthy humans [25,30], but that is proving to be a challenging endeavour as the taxonomic composition of the human microbiome varies between people [31].

While metagenomic sequencing does find many more species than culturing methods, it does have its own issues and biases. As there are not comprehensive reference genomes for every species, and creating reference genomes using cultured microbes cannot capture the natural variation found in the field, assembly and binning software do not always correctly reconstruct the genome. Repeat regions and sequences that have a significantly different GC content than the rest of the genome are frequently not binned by common software [32].

## 3. Curating Metagenomes

Data curation is essential when working with metagenomes sourced from online databanks. Curation in this context refers to collecting the data and sorting it into useable categories. Metagenomes can either be manually curated by the researchers looking to use the data, or by an algorithm trained to sort the metagenomes based on different features of the sequences.

Ontologies are systems for categorising data. Numerous ontologies designed for many different types of data can be found on the online repository www.bioontology.org (accessed on 1 April 2022) [33]. One ontology for categorising metagenomic data is the Functional Ontology Assignments for Metagenomes (FOAM) which uses Hidden Markov Models to classify gene functions [34]. Some other biological ontologies include the Biological Collections Ontology [35], or Interlinking Ontology for Biological Concepts [36]. One of the better-known ontologies for categorising biological samples is the environment ontology (ENVO). ENVO uses a directed acyclic graph (DAG) to categorise metagenomes based on the environment from which samples are sourced [37].

Because “environment” is a vaguely defined concept that is often made up of many smaller factors, ENVO uses multiple descriptors in its categorization process. For example, the surface of a stone at the bottom of a lake is both a rock and a lake environment. Depending on factors such as the depth of the lake and the water clarity, the microbiome of the rock surface may vary significantly. While the capability to factor in such details is important, categories can end up being split incredibly finely. In extreme cases, any ontology that attempts to account for all environmental details risks becoming overly complex and difficult to work with in terms of manual sorting of samples. Many ontologies avoid this by being more specialised for different research fields such as marine environments [38]. Training an algorithm to sort samples into the finer categories of an ontology would alleviate this difficulty. However, automatic curation has its own set of issues that need to be considered, as discussed below.

## 4. Issues with Curating Metagenomes

Manually curated genomes are susceptible to human error and rely on the data generator to provide sufficient and accurate information while uploading the sequences. As a result, manual curations often lack critical information and contain mistakes in the metadata provided.

Over the last few years, the number of metagenomes uploaded to databases such as the Sequence Read Archive (SRA) or MGnify has increased exponentially [39,40]. While this does provide more data for anyone to use the amounts of low-quality, contaminated, or mislabelled sequences have also increased. This adds to the amount of work required to manually curate enough genomes. While automated curation of metagenomes could address these issues, this approach requires that the categories have little overlap in the data being analysed by the algorithm, which can prove difficult. The datasets used to train such an algorithm would also need to be carefully curated to avoid contaminated or low-quality sequences, or sequences collected using very different methods. For example, in the SRA database, both amplicon sequencing and whole-genome shotgun sequencing can be categorised as metagenomic data [41]. This can be an issue as they provide different information that could cause issues for an algorithm trying to compare them. An algorithm can successfully differentiate between the two types of data; however, as the partition engine, PARTIE, has sorted many sequences from the SRA into amplicon and whole genome sequencing datasets [41]. Any attempts to automatically curate metagenomic data from the SRA database will need to take the broad definition of metagenomic data into account when collecting the training data.

## 5. Categorising Metagenomes

Distinguishing distinct environment categories for microbiome ontologies is difficult for a number of reasons. For human microbiomes, sequences vary greatly between individuals [42,43,44], between different locations on a single individual [43], and even within the same location on a single individual over time [42,43,45].

The metagenome of an external environment also changes over time [46] or in response to minor variations [47] in features such as water depth [5]. Furthermore, large variations within environments may warrant subdividing the environmental category further. For example, previous attempts to test the ability of algorithms to assign metagenomes to environments had high rates of incorrect assignments in some categories, particularly the “human respiratory” category [48]. This category included the lungs, mouth, nose, throat, and a few dental plaque samples. The large error makes sense, as both anatomically and biologically the mouth and the lungs are entirely different environments and cultivate different microbiomes [45].

Depending on how specific the ontology needs to be, separate categories for healthy and unhealthy sample environments should also be considered, as the microbiomes of healthy and unhealthy humans and animals can vary significantly [43,49,50]. However, unhealthy microbiomes can also blur the lines between closely linked environments as opportunistic microbes take the chance to colonise other environments. For example, *Rothia mucilaginosa*, a common oral microbe, has been shown to infect the lungs of cystic fibrosis patients [51,52].

## 6. Using AI to Curate Metagenomes

Most existing genomic databases do not require uploaders to provide comprehensive sample information (metadata) during the submission process. This results in many sequences being submitted to databases with little to no context. Manual curation often relies on this contextual information. The impacts of these missing details can be broad, as they translate into reducing the pool of sequences that can be used for other projects. For example, a water sample without metadata detailing the depth at which the sample was taken will be unable to be used to answer a question about the difference in microbiomes at various water depths as it cannot be assigned to any relevant category.

Artificial intelligence (AI) algorithms hold considerable promise in metagenomics, including for curation. For example, differences between metagenomes could be used to automatically and accurately curate based on the environment from which samples are isolated. However, this sort of decision-making activity will only happen once enough existing data is categorised and labelled appropriately.

It is also important to consider hierarchy of labelling: for example, how should the choice of labelling categories be managed for samples collected from different places on the same person? We previously used a random forest algorithm to sort metagenomes into categories based on their environmental source with 78.5% accuracy [48]. There is potential to improve that accuracy by either refining the training data or categories, and/or by changing the type of information the algorithm uses to curate the metagenomes.

Automated curation could radically change how researchers do microbiome research. Sequences that are unusable for large scale data integrations due to insufficient metadata could become available if a pipeline for automatic curation became available.

## 7. Functional Diversity vs. Taxonomic Diversity

Different types of data can be used for curation. For example, in previous work by this lab the taxonomic profile of metagenomes was used to curate them [48]. This approach is tempered because taxonomic make-up can vary greatly in different variations of the same environment. For example, the human microbiome varies greatly between people and can be influenced by diet [44], weight [42], medication [53], and many other factors [44,54].

Another approach to curation is to use the functional profile. These genes are often relatively conserved across microbiomes within similar environments, with more similarity in the relative abundance of functional genes compared to bacterial phyla in the microbiome of different environments [31,42,55]. It is possible a ‘core microbiome’ of functional genes that is similar across groups of environments may exist.

Overall, while taxonomic differences between people’s microbiomes can differentiate between samples on a personal scale, use of the functional profile for larger-scale metagenome curation could be more accurate. However, further research is required to compare the two curation methods to consolidate this hypothesis.

## 8. Phages and the Microbiome

Access to metagenomics has transformed how we study the microbiome, including research focusing on bacteriophages. Commonly referred to as phages, these viruses infect bacteria and are an important part of almost every microbiome. Individual phages often have a very limited host range, although almost every species of bacteria have phages that prey on them [56]. There are two main categories of phages, virulent phages, and temperate phages.

Virulent phages infect bacteria and hijack their replication machinery to make more copies of themselves, before releasing the new virions into the environment, often by lysing the bacterial host [57]. These lytic viruses are thought to kill approximately 20% of the ocean’s microbial biomass each day [58]. The targeted application of lytic phages to treat bacterial infections in humans and animals can be used as an alternative or to augment antibiotics and has seen recent success in compassionate-use cases [59,60,61,62,63,64,65]. Not all virulent phages kill their hosts however, some filamentous phages are extruded from their hosts without killing them and may help their bacterial hosts. For example, phage Pf4 helps its *Pseudomonas aeruginosa* host with biofilm maintenance [66,67].

Temperate phages switch between lytic and lysogenic lifecycles in response to environmental conditions and/or external stresses that induce the phage. When in a lysogenic cycle they either integrate into the bacterial host’s DNA and replicate with the host’s chromosome, or are maintained extrachromosomally as plasmids. Referred to as prophages, they can remain within hosts for thousands of generations [68].

Through lysogenic conversion, host bacteria can gain a variety of potentially beneficial traits that were encoding by the prophage’s DNA. For example, many prophages provide their hosts with protection against infection by other phages [69,70]. Prophages can also provide genes conferring resistance to antimicrobial drugs, protection from immune responses, or production of toxins [71,72,73,74]. For example, a prophage encodes the toxin genes that enable *Vibrio cholerae* to cause the life-threatening disease cholera [75]. The symbiotic relationship between phage and host can be parasitic as well as mutualistic: prophages can be a source of Selfish Genetic Elements which, as far as we know, only increase the replication of their own DNA and have a neutral or detrimental parasitic relationship with the host [76].

## 9. Piggyback-the-Winner and Other Hypotheses

The mechanisms that trigger the lytic-lysogenic switch are most well known in phage lambda [77], but various studies on the abundance of free temperate phages in various environments have led to a few hypotheses about general environmental conditions that influence the change in phage lifecycles. Three popular hypotheses are the Piggyback-the-Winner (PTW) [78], Piggyback-the-Loser (PTL) (aka refugium hypothesis), and Kill-the-Winner (KTW) hypotheses [79]. Each hypothesis is characterized by different environmental conditions (Table 1).

The Kill-the-Winner hypothesis is the oldest of the three and was first proposed in 2000 by Thingstad and looks at the problem from the perspective of the lytic lifecycle. KTW proposes that a lytic phage is more likely to come across a bacterium it can infect if that bacterium is more abundant. Therefore, abundant bacteria are encountering and being infected by lytic phages more often than bacteria that are not growing as robustly [79].

The Piggyback-the-Winner hypothesis was the next one proposed by Knowles et al. in 2016. In contrasts to the KTW hypothesis, it suggests that while host bacterial populations are replicating quickly, the phage uses a lysogenic lifecycle to take advantage of the host’s success [78]. This hypothesis also suggests the phage’s lytic lifecycle is triggered when bacteria start to become stressed. They supported their hypothesis from data from coral reef microbiomes and compared that data to previous models as well as their own hypothesis.

The third hypothesis is Piggyback-the-Loser (PTL) that suggests the opposite of the PTW hypothesis. The PTL hypothesis argues that when the host is growing slowly it is more beneficial to remain in a lysogenic state, as it is less likely that the phages produced through lysis will find a new host [80]. This hypothesis has also been called Piggyback-the-Persistent (PTP) [81] as the prophages were found in host species that persistently occurred at low abundance. PTP was first proposed in 2019 by Paterson et al. This study looked at the virus–microbe ratio in aquifer environments and found that the viral abundance was incredibly low. They were also one of the first groups to float the idea that the three hypotheses are competing strategies that become more or less prominent in different conditions [81].

Each of these hypotheses explains phage–host dynamics in studied environments. For example, PTL occurs in some regions of the deep sea or polar oceans [80,82] but PTW dynamics occur in some coral reef regions [83]. However, each hypothesis is an incomplete explanation of microbiome dynamics on its own. PTW and KTW also appear to be conflicting hypotheses (Figure 1) as they both have been shown to occur in high-energy environments but have the opposite results [79,80,83]. However, a recent study has taken a more in depth look at how the three hypotheses can coexist [80].

Silveira et al. propose that a combination of different mechanisms results in lysogeny being promoted at both very high and very low host abundances [80]. Their new hypothesis suggests that the effects of microbial density, community diversity, host–virus interactions, host metabolism, and the environment combine to either promote or repress lysis depending on which factors are more prominent. Overall, the main factors at play in determining whether a phage will be lytic or lysogenic are thought to be (1) chance of co-infections, and (2) host metabolism.

Co-infection is one of the best predictors for lysogenization [80] as it increases the abundance of phage repressors in the host cell, preventing lysis [84]. Slower host growth increases the window for co-infections to occur, which increases the potential for co-infections to promote lysogeny, supporting the PTL hypothesis. Conversely, hosts that are growing quickly have a shorter window for co-infection. However, eventually, the abundance of phages in the environment will increase to the point where co-infections happen quickly, leading to increased lysogeny at high host abundances which supports the PTW hypothesis. Co-infection is also affected by community diversity. When the community is less diverse phages are more likely to encounter bacteria that they can infect, which increases the probability of coinfections (compared to the lower probability of coming across an appropriate host if it is a relatively rare species).

Like co-infection, cell metabolism favours lysogeny at the extreme ends of the spectrum. In a low-energy environment where the hosts are growing slowly, cell starvation causes a signalling cascade. This increases the abundance of phage repressors [84], favouring lysogeny. When the hosts are not starving, they increase their stores of ATP. This allows the host proteases to degrade the phage repressors, and increase the chance of lysis [84]. However, in a very high-energy environment with limited oxygen, bacteria switch to a less efficient form of metabolism (e.g., fermentation) leading to low ATP/high NADPH levels. This prevents the host proteases from destroying the phage repressors and thereby limiting lysis.

As influencing factors, co-infection and metabolism work together to favour lysogeny at both very high and very low bacterial abundances. Silveira and colleagues suggest the rate of lysis peaks in environments with a bacterial concentration of 10^6^ per millilitre [80]. They propose that starvation and large windows for co-infection to occur favour lysogeny in low energy environments, while high bacterial concentrations and inefficient metabolism favour lysogeny in higher energy environments.

Silveira et al. suggests approximate bacterial concentrations for different virus–host ratios. In theory, this means that this hypothesis could be tested by counting the prophages, viruses, and bacteria in different environmental metagenomes and modelling environmental carrying capacity from the number of prophages in each sample [68]. A further test is to experimentally manipulate the virus-to-microbe ratio (VMR), for example by adding phages or removing bacteria (e.g., by filtration).

## 10. Prophage Abundances in Different Environments

Environments vary greatly in the amount of bacterial growth they can sustain, called the carrying capacity. Each situation exists at a unique point on Silveira et al. model [80]. For example, there are likely to be relatively few prophages per cell in low-energy environments such as the deep ocean, though likely still enough to trigger lysogeny, because prophages take resources to maintain. In higher energy environments—such as the ocean’s surface or a coral reef—lytic growth is also the dominant lifestyle, so very few prophages would be expected in bacterial genomes. In highly productive environments, such as coastal waters and estuaries, the environments favour lysogeny. However, here, the energy cost to maintain the prophages is not as big of a limiting factor (because energy is plentiful), so the number of prophages per cell should theoretically be much greater than in low energy environments.

Numerous studies have examined phages across many different environments, including in various animals [85], in humans [86,87,88,89], from most aquatic environments [71,90,91,92,93], and from soil [94,95,96]. While many of these papers examine phages from single environments, there are comparatively few studies that compare phages from different environments in one analysis. Of these, many only compare a couple of environments at once [71,89,93,97]. As different prophage detection methods can produce slightly different results [98], comparing phages from environmental samples that have been collected and analysed with various methods could produce inaccurate results.

## 11. Identifying Prophages

The first step in analysing prophages is to identify their sequences in the bacterial genome. The approaches for doing so broadly separate into two main categories: experimental induction, or computational examination of a bacterial genome for prophage-like regions.

Induction involves causing lysis in the bacteria, which releases the phage into solution. It is one of the oldest methods for detecting phages and uses chemicals that signal to the phage to switch to a lytic lifecycle, destroying the host bacteria and releasing the phages into the solution. The issue with this method is that it can only be used on culturable bacteria and incomplete induction will cause the phage-host ratio to be skewed. Although we have discovered many chemical inducers, there are also plenty of prophages where we have not identified the signals they are sensing, and so cannot induce them.

Computational examination of bacterial genomes first requires that the bacterial DNA is sequenced, then the sequences can be examined for genes that belong to prophages either manually or by using programs such as PhiSpy [99] and many others [98]. The accuracy of these predictions is affected by the quality of the sequence assembly. As genomes are being assembled small, overlapping segments of DNA sequences are pieced together to form larger sequences, or contigs. A genome with fewer contigs is more complete than one with more contigs. A genome with many contigs generally has a lot more predicted prophages than genomes with few contigs and the rate of false positives increases as the number of contigs in a genome increases as it is more likely for a smaller segment of DNA to appear the same as a prophage by chance alone. A comparison of eight different prophage prediction tools showed that each package has its strengths and weaknesses [98]. Some sacrificed speed and efficiency for accuracy while others tried to find a balance. Some had good precision but that necessitates an increased chance of false-positive results, while others increased recall performance which comes with more false positives. The pros and cons of each program need to be evaluated by users before deciding which is most appropriate for each project.

## 12. Conclusions

Metagenomic sequencing is useful for finding prophages from many different environmental conditions, but many genomes are added to databases without the inclusion of comprehensive metadata. Being able to automatically sort these sequences into an environmental ontology would allow for these sequences to be useful in future projects, but we need considerably more high-quality data to determine how best to sort these sequences.

Prophages play significant roles in the microbiomes of many species and in different environments. Phages can protect their host from deadly infections, and give their host access to beneficial genes such as antimicrobial resistance or toxin production. The way phages interact with their host changes depending on the environment. Researchers have conducted many studies on phages from various environments, and developed hypotheses regarding what factors influence survival strategies, such as the lytic/lysogenic decision. However, there is still much more to learn about how prophages interact with their hosts under different conditions.

Learning more about metagenomes and prophages could provide many insights into human and environmental health, while obtaining a better understanding of what a healthy microbiome should be may enable us to detect changes more quickly or accurately in microbiomes that could be a sign of disease.

## Figures and Tables

**Figure 1 microorganisms-10-01671-f001:**
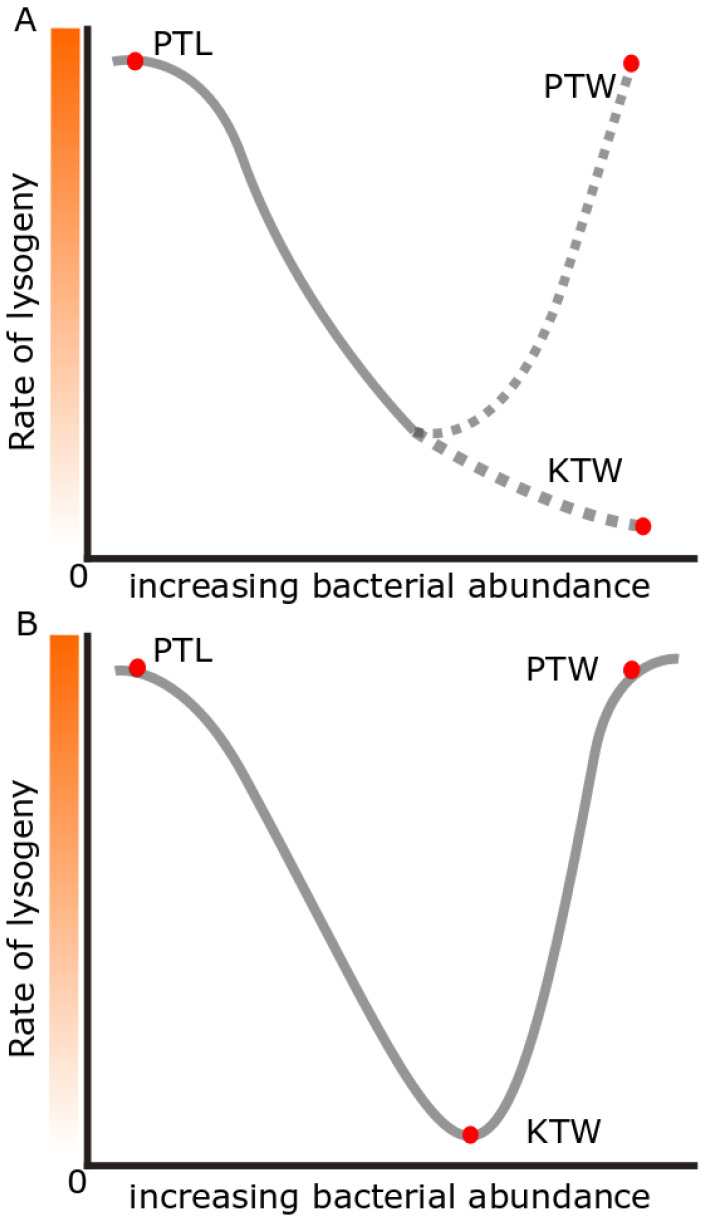
Plotting the hypothetical rates of lysogeny between the PTL and KTW/PTW dynamics shows how conflicting the hypotheses appear to be (**A**). However, a recent hypothesis [80] suggests that the previous hypotheses can coexist and PTW overtakes KTW dynamics at high enough bacterial abundances (**B**).

**Table 1 microorganisms-10-01671-t001:** A brief summary of the different hypotheses for explaining lysogeny rates. Both Piggyback-the-Winner (PTW) and Kill-the-Winner hypotheses (KTW) occur in similar environmental conditions.

Lysogeny Hypothesis	Environmental Conditions	Level of Lysogeny
Kill-the-Winner	High energy, cells growing	Low
Piggyback-the-Winner	High energy, cells growing	High
Piggyback-the-Loser	Low energy, little growth	High

## Data Availability

Not applicable.

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
