# Peer review of "How Metagenomics Has Transformed Our Understanding of Bacteriophages in Microbiome Research"

_microorganisms, 2022, doi:10.3390/microorganisms10081671_

Round 1
Reviewer 1 Report
The authors aimed to present an interesting perspective on metagenomics and how the field has enabled advancing the understanding of bacteriophages.
Overall, the paper writing style is not well presented. It is hard to follow an idea because there are punctuation marks that are needed.
In addition, tThe overall ideas have been presented previously in different publications. For these reasons, the manuscript may not be suitable for publication.
Reviewer 2 Report
The authors in this article aim at reviewing how metagenomics benefited to the understanding of bacteriophages in microbiomes. They summarized the history of metagenomics in a concise manner. Their introduction of bacteriophages lacks some precisions and even display some confusing statements. Key concepts are either missing or spread in different paragraphs. The authors should regroup all needed information about phages in one single paragraph (see below some comments and suggestions). The authors do not insist enough on the difficulties one can encounter to reconstruct and sort genomes from metagenomic data between bacterial sequences, virulent and temperate phages. Also, I feel there is a need of more illustrative cases of studies that did improve our understanding of bacteriophages (both temperate and virulent) contributions to the life of a microbiota.
Everything considered, I recommend publication of this review pending improvements both in the required information to better understand phages and more illustrative cases.
Concepts of lytic and lysogenic cycles.
Lines 203-205/220-221 introduce the two major bacteriophage lifestyles one can observe in host/virus relationship. Some key concepts must be introduced more clearly to avoid confusion. In lines 203-205, the authors state that phages occupy either a predatory or a symbiotic role. I am not at ease with "predatory" and "symbiotic" qualifiers.
Predatory
Bacteriophages (and viruses in general) are first of all parasitic biological entities: they need at least the host translational machinery to produce new virions. The dependency on host's replication machineries varies between phage species. Some goes as far as coding for their own RNA polymerase while others completely depend on the host replication and transcription enzymes.
One can describe two main phage categories: the virulent (not "lytic", replace all instances in the manuscript when necessary) and the temperate phages. They differ from the life-styles they can adopt: the lytic (here "lytic" is correct) and the lysogenic cycles. Virulent phages only follow the lytic cycle while temperate phages can adopt both lytic and lysogenic cycles, depending on environmental conditions.
I do understand - and in a way approve - the concept of predation applied to phages as most of the known viruses trigger cell lysis. But it is not always the case: some filamentous phages – while being virulent phages - are actively extruded from the cell without killing its host. In some instances, phage infection is also chronic. For these reasons, I would be a bit cautious with the term "predatory" by adding a sentence to cover virulent phages that do not kill their host or chronic infection.
Symbiotic
My opinion is that the concept of symbiosis for temperate phages integrated into its host's genome as a prophage is more debatable. Through lysogenic conversion (a term that should appear in the manuscript), the host can benefit from prophage-encoded functions. In addition to the benefits mentioned by the authors lines 213-218, I would add protection against infection by other phages. But it is not always the case that the host seems to benefit from its prophage. There are some other mobile genetic elements that are described as "selfish genetic elements" that do not seem to confer any advantages to the host (or at least at the present state of our knowledge).
Temperate phages
The authors should be cautious with the prediction of temperate phages as they rely on the sole, well-known and characterized Lambda phage where decision-making between lytic and lysogenic cycle is well described. But this mechanism is not universal to all temperate phages. For other temperate phages that do not resemble Lambda, little is known.
Line 220
All prophages are temperate phages (or used to be if they are not able anymore the resume the lytic cycle). To be more precise with the phage-related vocabulary I would replace the sentence line 221 – 222 " (…) when conditions change." by " (…) when environmental conditions and/or external stresses induce the prophage." For instance, addition of DNA-damaging drugs to the culture such as Mytomycin C is a classical way to induce prophages via the activation of the bacterial SOS response (but other stresses may serve). This is somehow described but later on in the manuscript and should then be regrouped in a single paragraph introducing bacteriophages.
Minor modifications
Line 50: replace "E. coli" by "Escherichia coli"
Line 176-177: reference
Line 183: is it a subtitle?
Line 218: "Vibrio cholerae" in italic.

Reviewer 3 Report
The current manuscript by Inglis and Edwards reviews the state of the art of the role of bacteriophage in the microbiome. It outlines the history of metagenomics and the insights gained through exploring how bacteriophages shape bacterial communities.
This is a comprehensive review that touches on all the important points related to biology and bioinformatics. I enjoy reading it. That said, there are still a couple of points that need to be addressed.
Major
In the part regarding to 'Categorization of metagenomes', it is stated that the mouth and the lungs are completely different environments. Here, it should be clarified that in disease-related microbiomes (i.e., CF) metagenomes colonizing the lungs have been found to be similar to those present in the mouth measured in sputum.
In the examples mentioned in the manuscript about the ability of phages to profile bacterial communities in ecosystems, it is also worth mentioning that phages can be used in the clinic as compassionate therapy. Currently, the use of phage is a hot topic due to its use as a therapy for chronic infections as an alternative to antibiotic resistant infections.
Minor
Please insert the acronyms in the main text before you start using them.
Round 2
Reviewer 2 Report
The authors have addressed the major points raised in my initial report.
Author Response
"The authors have addressed the major points raised in my initial report."
RESPONSE: Thank you for your feedback.
Reviewer 3 Report
The current version of the manuscript is ready for publications. Thanks to the authors for address my comments.
A minor change:
Line 98: "aren't" for "are not"
Author Response
The current version of the manuscript is ready for publications. Thanks to the authors for address my comments.
A minor change:
Line 98: "aren't" for "are not"
RESPONSE: Thank you for pointing that out. I have fixed the error.